# “Cow Healers Use It for Both Horses and Cattle”: The Rise and Fall of the Ethnoveterinary Use of *Peucedanum ostruthium* (L.) Koch (fam. Apiaceae) in Sweden

**DOI:** 10.3390/plants12010116

**Published:** 2022-12-26

**Authors:** Erik de Vahl, Giulia Mattalia, Ingvar Svanberg

**Affiliations:** 1Department of Landscape Architecture, Planning and Management, Swedish Agricultural University, POM, SE-23422 Lomma, Sweden; 2Institute for Russian and Eurasian Studies, Uppsala University, SE-75120 Uppsala, Sweden; 3Department of Environmental Sciences, Informatics and Statistics, Università Ca’ Foscari Venezia, 30123 Venice, Italy

**Keywords:** cultural relict plants, herbal remedies, historical ethnobotany, living biocultural heritage, silvopastoral system

## Abstract

Masterwort, *Peucedanum ostruthium* (L.) Koch, is an Apiaceae species originally native to the mountain areas of central and southern Europe. Written sources show that it was used in northern Europe. This study explores the cultivation history of masterwort and its past use in Sweden. Although only few details are known about the history of this taxon, it represents a cultural relict plant of an intentionally introduced species known in Sweden as early as the Middle Ages. In Sweden, the masterwort was mainly used as an ethnoveterinary herbal remedy from the seventeenth to nineteenth centuries. However, medicinal manuals, pharmacopoeias and some ethnographical records indicate that it was once also used in remedies for humans. Today, this species remains as a living biocultural heritage in rural areas, especially on the surviving shielings, which were once used as mountain pastures in Dalecarlia, and at former crofts that were inhabited by cattle owners in the forest areas of southern Sweden.

## 1. Introduction

### 1.1. Background

Between 2007 and 2017, a nationwide inventory organised by the Programme for Diversity of Cultivated Plants (POM) collected data and living material for cultural heritage species in Sweden. The plant material was documented together with information regarding its cultivation history, use and traditions [1]. Many interesting landraces and old cultivars of herbs, vegetables and ornamental plants were found [2,3,4]. In the POM inventory, a specimen of masterwort, *Peucedanum ostruthium* (L.) Koch, synonym *Imperatoria ostruthium* L. (fam. Apiaceae), was found in Leksand parish in the province of Dalecarlia, central Sweden. This taxon has survived as a cultural relict plant for centuries, especially in Dalecarlia, but also in some other areas of Sweden [5]. 

When the young Carl Linnaeus and his travel companions passed by Nås parish in western Dalecarlia in the rainy summer of 1734, he noted in his diary on 11 August that *P. ostruthium* was cultivated there [6]. Later, Linnaeus also mentioned its presence in the mountainous areas of Lima parish (i.e., in the Transtrand area) in the same province [7]. Linnaeus’s earlier observations on the cultivation of masterwort at the location were confirmed when topographer and writer Abraham Hülphers described Lima parish in 1762 [8]. Hülphers wrote about the importance of cultivating medicinal plants in such a remote and isolated area, where the vicar also served as the local physician. *P. ostruthium* was reported to be used with hard liquor (Swedish: *brännvin*) against colic, and as a smoked substance for calming nosebleeds, ear pain and toothache [8]. Currently, the use of *P. ostruthium* is reported as a medicinal plant for human and veterinary purposes by several ethnobotanical studies in the Alpine region (i.e., in Austria [9,10], Italy [11] and Switzerland [12] ) (Figure 1).

### 1.2. Biology

Masterwort is a hemicryptophyte and stem plant. It is a perennial herb with large rhizomes and a round stalk that grows 30 to 100 cm high. The stalk is erect, hollow, round, leafy and slightly branched. Lower leaves are on long stalks, twice ternate. The upper leaves are less compound and on shorter stalks, with a sheeting, membranous dilatation at the base. The flowers are white and slightly reddish, and sit in a pair of large, somewhat flat umbels. The plant does not bloom every year, but propagation occurs vegetatively through underground runners [13]. 

The rhizomes contain between 0.18 and 0.78 percent essential oils, especially sabinene. The plant is a source of coumarins, including 1.3 percent oxypeucedanin (C_13_H_12_O_2_), 0.3 percent ostruthol (C_24_H_24_O_8_), imperatorin, 0.1 percent osthole (C_12_H_18_O_2)_, isoimperatorin and 0.5 percent ostruthin (C_18_H_20_O_8_) [10,14,15]. 

### 1.3. Distribution and Ecology

*P. ostruthium* grows wild at altitudes between 800 m and 2800 m in the mountains of central and southern Europe, in meadows and along streams. Its native distribution includes the Carpathians, the Alps, the northern Apennines, the Massif Central, the Jura Mountains and scattered occurrences in the Iberian Peninsula. In Germany, it is found in the Harz area, the Thuringia and the Ore Mountains (Erzgebirge). Since it is widely introduced and cultivated as a medicinal plant, its native range is not entirely clear [9,13,16]. In the British Isles (northern England, mid to northern Scotland, Isle of Man and northwest Ireland), it is considered introduced and naturalised in moist meadows and riverbanks [17]. In the Scandinavian countries, it has been introduced in Denmark, Norway and Sweden. It reached Scandinavia in the late medieval times [18,19,20].

From modern provincial floras, it appears to be found as a cultural relict plant in forest areas from Skåne up to Dalecarlia, where it can survive as a reminder of older times [21]. In Dalecarlia, it was reported from 140 recording sites in 32 parishes until 1960, especially at shielings. However, it has also been found in abandoned places of residence in the lowland areas of Dalecarlia [22]. SLU Swedish Species Information Centre reports known observations (until 2022) from the provinces of Skåne (20), Blekinge (9), Småland (422), Halland (68), Bohuslän (86), Gotland (8), Öland (3), Västergötland (361), Östergötland (191), Dalsland (68), Södermanland (24), Värmland (25), Närke (56), Västmanland (60), Dalecarlia (302), Gästrikland (47), Hälsingland (22), Härjedalen (2), Jämtland (4), Åsele lappmark (1), Pite lappmark (14), Lycksele lappmark (1) and Västerbotten (7) (Figure 2) [23]. 

It can survive for a long time in old yards if there is not too much competition with other plants or from forests. *P. ostruthium* does not flower very often, and the flowers are rarely fully formed. It is highly uncertain whether the plant can form viable seeds in Sweden [21].

Today, the plant is rarely cultivated in Sweden, other than at botanical gardens (e.g., Uppsala Botanical Garden, the Friends of the Garden Society garden in Vadstena, Floras Garden in Sollentuna, Vallby open air-museum in Västerås and Svanå Garden in Boden, according to our knowledge). It is also preserved at the Swedish National Gene Bank for Vegetatively Propagated Horticultural Crops at the Swedish Agricultural University in Alnarp. 

### 1.4. Aim of the Article

The main aim of this study is to bring together historical and contemporary data about *P. ostruthium* and to describe its earlier importance as a medicinal plant in Sweden. Specifically, we document its introduction and cultivation history in Sweden and its folk botanical importance, especially within ethnoveterinary medicine, among the pre-industrial cattle-breeding peasantry.

## 2. Material and Methods

### Data Collection and Analysis

Older kitchen herbs, vegetables and medicinal plants were searched for when producing POM’s nationwide inventories of cultivated plants [2,3,4]. Only four notifications regarding *P. ostruthium* were received, all of which were collected for cultivation trials and conservation [2]. In addition to POM’s plant inventories, botanical handbooks, ethnographic archive materials and local historical works have also been used as sources [24]. A diachronic perspective was chosen in order to outline and analyse the regression and changes in the use of masterwort, during the course of which we took into account the social, ecological and chemical aspects of this plant’s usage [4,24].

## 3. Results

### 3.1. Ethnographic Context

Several sources agree on linking masterwort to silvopastoral activities. For instance, in the eighteenth and nineteenth centuries, travellers and botanists in Dalecarlia noticed that *P. ostruthium* was growing at the shielings (Swedish: *fäbodar*) used by peasants as mountain pastures for grazing their livestock in the summertime [25,26]. The female herders cultivated the plant in order to use it for ethnoveterinary purposes [27]. However, it is also known to had been grown by cattle-keeping crofters in forest areas of central and southern parts of Sweden [21,26]. 

One specific biocultural context for the species is the transhumance pasturing system used in the province of Dalecarlia. From June until the end of September, the young peasant women would spend the summer in the shielings up in the forest-covered mountains, tending the animals and making various dairy products, while the men spent their days further away, fishing and haymaking in the wet meadows [27,28]. Although most of these shielings were abandoned in the mid-twentieth century or earlier, a few remain still active. The remaining shielings have now been converted into summerhouses. The lifestyle and the human–animal relationship associated with the shielings included many noteworthy cultural traits of great interest for ethnobiologists, for instance the special kind of song form (*kulning*) used by the women as a way to communicate not only with each other, but also with the cattle [29]. The material culture adapted to the herding way of life is also of great interest. While spending their days in the forests, the girls and women also gathered local resources which they could sell after returning to the villages on Michaelmas (29 September) [30]. These products included bundles of *Equisetum hyemale* L., which were used to scour or clean wooden milk vessels, and lumps of resin from *Picea abies* (L.) H. Karst, used as a kind of ‘chewing gum’ by the peasantry during church services [27]. Few plants were cultivated at the shielings, although some herbaceous taxa used as veterinary herbs were grown, for instance lovage (*Levisticum officinale* L.), tansy (*Tanacetum vulgare* (L.) Bernh.) and the previously mentioned masterwort [31]. The species has also been cultivated by livestock farmers in southern and central Sweden [29,32,33]. In some areas, horseheal (*Inula helenium* L.) has also been grown as a livestock medicine [34,35].

The relationship between humans and *P. ostruthium* dates back to medieval times in Sweden. However, this interaction has changed over time. Some other plants, like *L. officinale* and *T. vulgare*, were also planted in order to control diseases among livestock. Today, these taxa remain as cultural relicts on the surviving shielings [25]. These plants have been cultivated for centuries, probably as early as the medieval times in Sweden [20]. *L. officinale* is still used as a vegetable and for seasoning, and is therefore cultivated in many Swedish gardens [36], while the use of *P. ostruthium* has mostly fallen into oblivion. Remaining *T. vulgare* is regarded as an ornamental plant at best [37]. *I. helenium* might be naturalised in some areas [35]. 

### 3.2. Cultivation History

The medieval Latin name *Imperatoria* (from *imperatoris*, meaning ‘ruler’ or ‘master’) was translated into French as *impératoire*, and rendered in English as *masterwort* (first recorded in 1653), in German as *Meisterwurz* (recorded since 1480), in Danish as *mesterrod* (recorded since 1678) and in Swedish as *mästerrot* (recorded since 1632), alluding to its reputation as a plant with superior healing properties [38]. ‘Master’ was once a title for a physician, and the plant was known by these names because it was regarded as a divine medicine [21,32,38]. 

Very little is known about its earlier history as a cultivated plant. However, it seems that in ancient times it was primarily cultivated for human ailments. Archaeological finds show that it was introduced into the British Isles by the tenth century. Seeds dated around 850 to 950 CE have been found in Antrim, Northern Ireland [39]. Swiss natural historian Conrad Gessner describes it as a cultivated plant in 1560 [38].

### 3.3. Medicinal Plant for Curing Humans

This taxon seems to have entered medicine as recently as the Middle Ages, probably in Germanic territory. Evidence from antiquity is lacking. The use of *P. ostruthium* in human remedies was first recorded in medieval times. A review of herbal books and medicinal manuals from the late Middle Ages shows that it was known as *Ostruhium, Astrantia and Magistrantia* [38]. In a Danish manuscript by the thirteenth-century Roskilde Cathedral canon Henrik Harpestraeng (who died in 1244), it is described as a kind of panacea used in remedies for liver diseases, jaundice, gallstones and cough. It is also mentioned as being grown in Danish physic gardens in the 1530s [40]. Italian physician Pietro Andrea Mattioli propagated it for its usefulness in making remedies in sixteenth century. So did German botanist Jakob Tabernaemontanus in the same century. English herbalist John Gerard wrote in his herbal 1597 that “the rotes and leaves stumped, doth dissolve and all pestilential carbunchles and botches, and such other apostemetions and swellings” [38] (Figure 3).

The canon Christiern Pedersen in Lund wrote in 1533 that it was grown in physic gardens, and recommended it for treating lower back pain [41]. Henrick Smid, who practiced medicine in Malmö, confirms in his herbal its use in human medicine [42]. 

In Swedish medicinal handbooks from the fifteenth and early sixteenth centuries, it is known as *Astrice*. The plant was recommended by Laurentius Gothus Paulinus in 1623 as a remedy against pestilence [43]. Nyström’s review of plants grown in apothecary gardens in Sweden during the eighteenth century confirms that the masterwort was among the species appearing in all gardens studied [44]. Apothecary gardens emerged due to the professionalisation of the art of medicine that developed during the century, and the species in cultivation did not always correspond to the plants used in folk medicine. *P. ostruthium* was included in the Swedish pharmacopoeia from 1698 (as *Radix imperatoriae*), but was withdrawn in 1869 [45,46]. However, it had fallen out of use in academic medicine much earlier. 

It did, however, continue to have some use in folk medicine [38]. In 1806, Retzius mentioned that the peasantry dried rhizomes for use against colic and mental illness in women, and it was given to small children twice a day in powder form to prevent intestinal worms [47]. We have some fragmentary data regarding its continued use for treating livestock (Upper Dalecarlia and Norrbotten) in the late nineteenth century [26,27]. Nowadays, it is forgotten in folk medicine. It is known among some current practitioners of alternative medicine, but we have no indication of whether it is actually used. Its supposed medicinal properties have also been claimed by mountain dwellers elsewhere in Europe, even up to the present day. In the Alpine regions of Europe, its rhizomes have been used for a great number of ailments, including gastrointestinal complaints, wounds, skin problems and toothache [48,49,50]. Among the Saami people in northern Sweden, the sharp tasting roots of a related wild species, milk parsley, *Peucedanum palustre* L., was earlier harvested and used against various ailments [51]. 

### 3.4. Ethnoveterinary Medicine

Despite having largely ceased to be used for treating humans in academic and folk medicine, it began to be cultivated in the eighteenth century for ethnoveterinary purposes by the peasantry in southern Sweden, as far north as the province of Dalecarlia [52,53]. It was also occasionally grown for ethnoveterinary purposes in the northern part of Sweden [33]. In POM’s nationwide surveys, the four accessions of masterwort described were given cultivar names relating to either the person who had cultivated them or how they were used [2]. Two of these, labelled ‘Brita-Sofia’ and ‘Skogen’, were described as surviving relict plants from old private gardens that could be dated through the living memories of the donors. However, they had no connection with any known medicinal use. The other two, known as ‘Kampkur’ (‘horse cure’) and ‘Kobota’ (‘cow healer’), bear witness to their use in folk veterinary medicine [2] (Figure 4).

‘Kampkur’ was collected from Rällsjögården in Bjursås parish, Dalecarlia, where the farmer Rällsjö Anders Jansson (1863–1951) was active as a folk healer at the turn of the twentieth century. Inspired by the book *Pharmaca Composita*, published in 1896 [54], Rällsjö Anders used the medicinal plants that had long been grown at the family’s shieling in Axmor for home remedies. The masterwort, called ‘Kampkur’, was included in a medicine for horses that also included rhubarb (*Rheum* sp.), mezereum (*Daphne mezereum* L.) and garden angelica (*Angelica archangelica* L.), and was preserved at the summer pasture by his daughter Rällsjö Brita (1901–2006), who had realised the value of maintaining both plants and memories from an older garden culture [55] (Figure 5).

‘Kobota’ was collected from Rosa Backman’s garden in Orust in Bohuslän, western Sweden. It bears the name Mrs Backman gave it to describe how it was traditionally used in the area. In the spring, when the cows were weak, leaves from *P. ostruthium* were gathered, rolled in rye flour and then stuffed down the animals’ throats. Mrs Backman recalls that the animals got up after a number of treatments, and she explained the effect of the plant with its high vitamin content. 

No chemical analyses of the four collections have been carried out to date, but sensory tests, made by E.D.V. at Alnarp, have shown that the chemical content of the roots differs greatly in terms of acidity, fruitiness, bitterness and aroma. They cannot be morphologically separated, but differences in vitality and plant vigour indicate a genetic variation within the collected and preserved plant material.

These ethnoveterinary uses of *P. ostruthium* are also confirmed by the cultural historical and ethnographic data. It was known and used against animal diseases in the early seventeenth century. Masterwort is mentioned in Mårten Behms’s 1648 medicinal handbook for treating horse diseases [56]. Åke Claesson Rålamb also mentioned it as remedy plant for horses, cattle and other domestic animals in his medicinal handbook dated 1690 [57]. It has been widely used against livestock diseases in particular, and was used as a diuretic and laxative for horses and against swine fever and rinderpest [58]. Still in the nineteenth century, sick swines were washed with decoction of masterwort in Dalecarlia [59]. 

Its use became popular in ethnoveterinary medicine in the eighteenth century. It was mainly the rhizomes that were used: fresh or dried, cut into pieces, taken as a decoction or chewed. The leaves were also used as treatments. Retzius summarised in 1806 how cattle healers ‘used it for both horses and cattle’, for example when treating loss of appetite [47]. Its use is recorded at many shielings [60,61]. The same use is also mentioned in other parts of Sweden [25,33]. Detailed data from Vilhelmina in Lapland describes how a local healer named Hans Persson in Latikberg used *P. ostruthium* to cure horses and cows [62]. 

Masterwort was also used as cattle medicine in other Northern European countries, for instance at the shielings in Norway. Høeg recorded that it was planted for use as a cow medicine, and that the herding women came from other shielings in the vicinity to harvest leaves to be used as medication [63]. Its use for ethnoveterinary purposes is recorded in Germany, Austria, Switzerland and northern Italy [64]. A record from Danish West Jutland states that masterwort was a common home remedy against “destruction by evil forces” in both animal and human folk medicine [40]. The addition of chopped masterwort and flax seed to beer, or sprinkled over the patient’s head, was claimed to protect against the devil [65]. 

## 4. Discussion

Our knowledge of plants used in ethnoveterinary medicine by the peasantry in pre-industrial Sweden is still limited [66]. POM’s inventories of cultural plants all over the country show that *P. ostruthium* was an important plant in folk veterinary medicine, and this has been a factor for finding and evaluating old plant material for long-term conservation. However, we can also learn that medicinal plants that have lost their function and place in modern gardens were sometimes neglected, and were only collected from donors with knowledge of old traditions. 

Sweden’s Virtual Herbarium includes records of masterwort from the areas around Lima [67]. All of these are from the early twentieth century. Reported localities in the Swedish Species Observation System (Artportalen) database show that the species was still growing at shielings in mountainous areas in the late 1980s. This indicates that the species has been growing continuously in the area [68]. Nevertheless, available sources do not mention how the plant was propagated between different shielings, while this information is reported for many other plant species in the inventory. We cannot tell why this plant was common among Dalecarlian pastoralists, but not in the regions north of Dalecarlia where shieling culture was also common.

POM’s inventories of heirloom plants have been based on information from the public and trained volunteers working in the field, and the method of selection was based on opportunities to link knowledge of the cultivation history of a plant to living memories or documentation of older horticultural culture [4]. This method has probably excluded species that have lost their importance, and in those cases where masterwort has been found, the donors’ knowledge of the plant’s historical use has been decisive. A different method based on information from the literature, such as that describing masterwort grown in Lima, would probably give a partially different result, but would also bring benefits in terms of securing the provenience of the plant material. Older herbarium records are a source material that can be important in such cases for establishing links between relict plants and historical cultivation.

The preservation value of masterwort plant material might not be as obvious as for other previously important medical plants that now have new functions in horticulture. For example, the social values of southernwood (*Artemisia abrotanum* L.) are linked to practices and memories of human interaction [3]. Multifunctional species such as tansy, *T. vulgare*, have survived in horticulture as ornamental plants, while lovage, *L. officinale*, is now used as a herb. Other former medical plants, such as rhubarb (*Rheum* sp.), have instead achieved new functions as vegetables, while knowledge of the species’ medicinal qualities has now been forgotten. Masterwort has lost its functions in Swedish gardens and practices, first as a pharmaceutical drug and then also in ethnoveterinary medicine due to traditional livestock breeding being abandoned and the modernisation of veterinary medicine [69]. This is evident in the more recent (and sometimes current) use of this plant for dietary, medicinal and veterinary purposes in other sociocultural and geographical contexts, such as Alpine pastures, where pastoralism is still practised. Indeed, *P. ostruthium* prefers nitrogen rich soil which is a characteristic of grazed pastures, especially those where cattle and herds stop for a while (e.g., for the night), or in the immediate proximity of shielings or mountain huts. Thus, the conservation of this species (and biodiversity more generally) may be fostered by pastoral activities [70], and it may also be used as a medicinal resource (Figure 6).

Masterwort has yet not gained popularity as an ornamental plant and is seldom included in reconstructed herb gardens. However, preserving the genetic spread of relict plant material, recording their use and documenting ethnobotanical knowledge are important tasks of the Swedish Programme for Diversity of Cultivated Plants in relation to the *Global Plan of Action for the conservation and sustainable utilization of plant genetic resources for food and agriculture, developed by FAO*. Collecting additional plant material from northern localities in Sweden, such as Lima, might be important in terms of contributing to the gene pool of the species from a global perspective [71]. 

## 5. Conclusion

A deeper understanding of traditional practices in connection with species commonly labelled as medicinal herbs might be important when reconstructing historical landscapes and gardens. The reconstruction of an eighteenth-century physic garden differs from the cultivation at shielings connected to silvopastoral systems, even though the same species are sometimes likely to be included. Both the preserved plant material and knowledge of folk medicine and ethnoveterinary uses are crucial in order to understand and preserve biocultural values linked to different cultural heritage elements.

## Figures and Tables

**Figure 1 plants-12-00116-f001:**
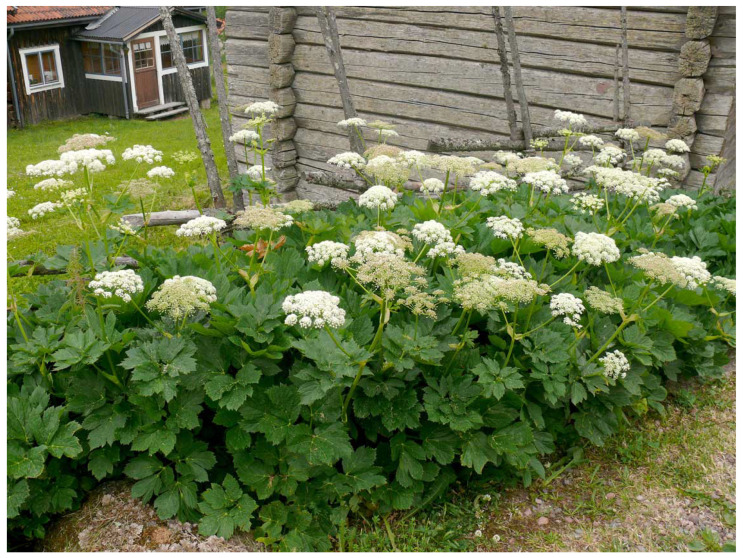
*Peucedanum ostruthium* in Orsa parish, Dalecarlia (Photo Arne Holmer).

**Figure 2 plants-12-00116-f002:**
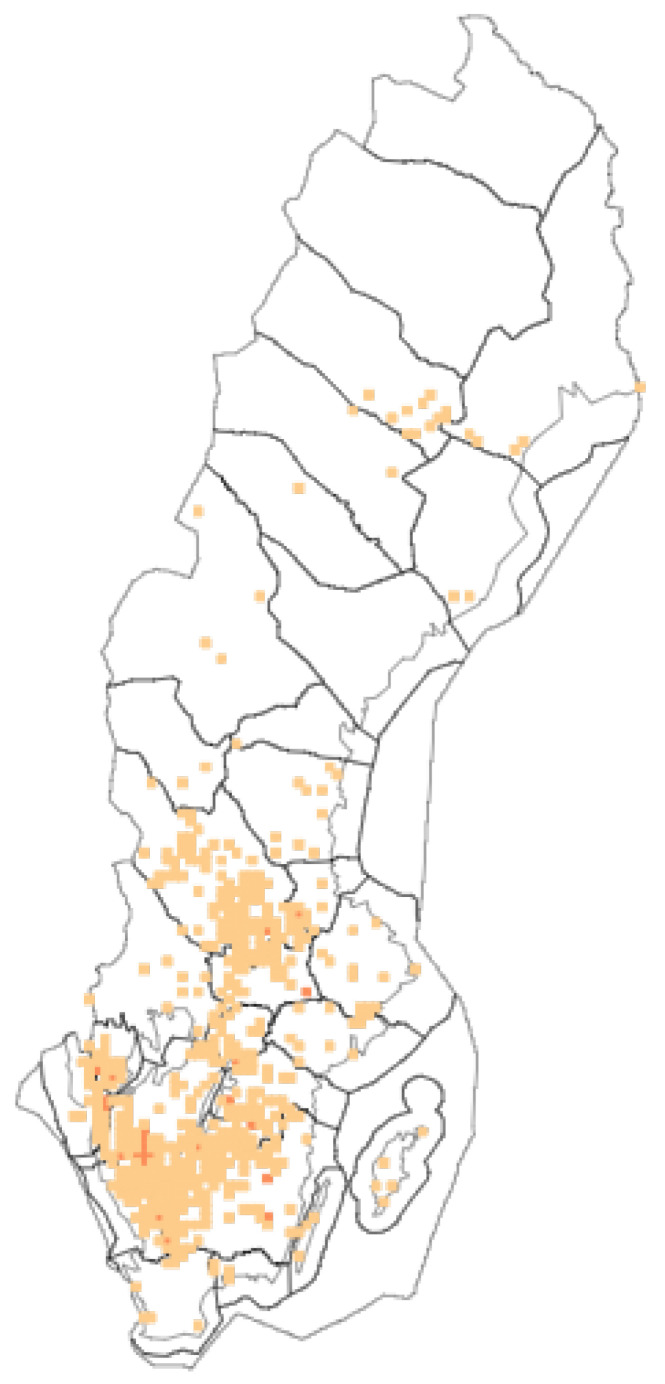
Contemporary distribution of *Peucedanum ostruthium* in Sweden. Source: SLU Swedish Species Information Centre. Light squares < 10 observations, dark squares 10–500 observations.

**Figure 3 plants-12-00116-f003:**
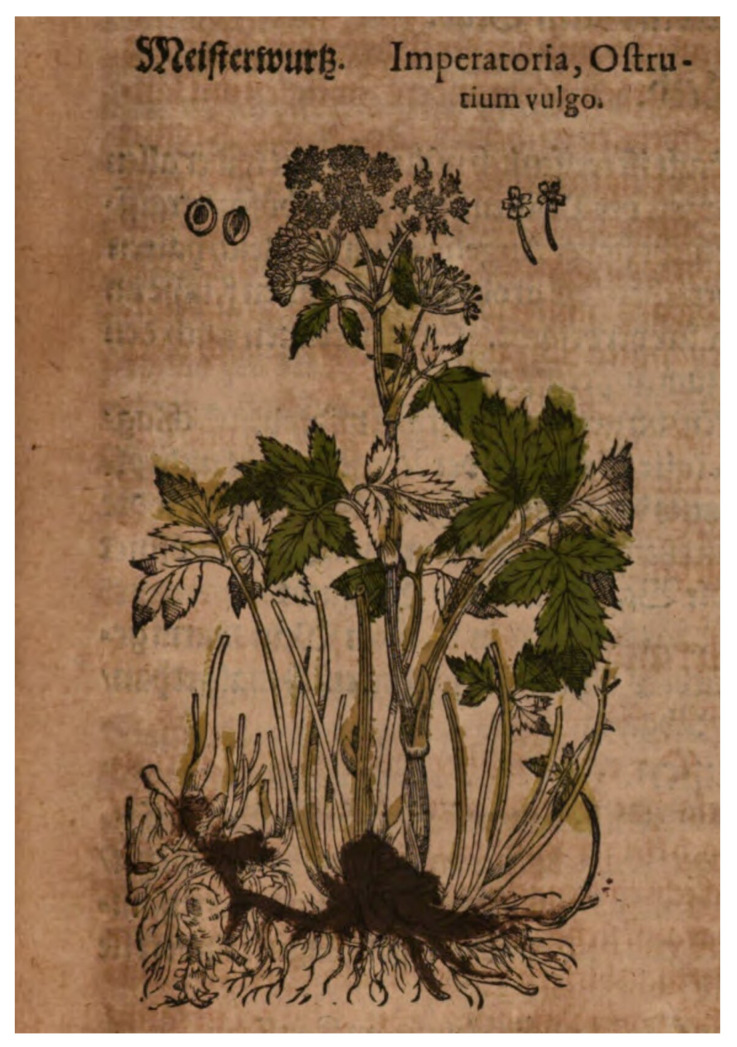
Masterwort in a colourised illustration from Pietro Andrea Mattioli’s *De plantis epitome utilissima*, published by Joachim Camerarius in 1611.

**Figure 4 plants-12-00116-f004:**
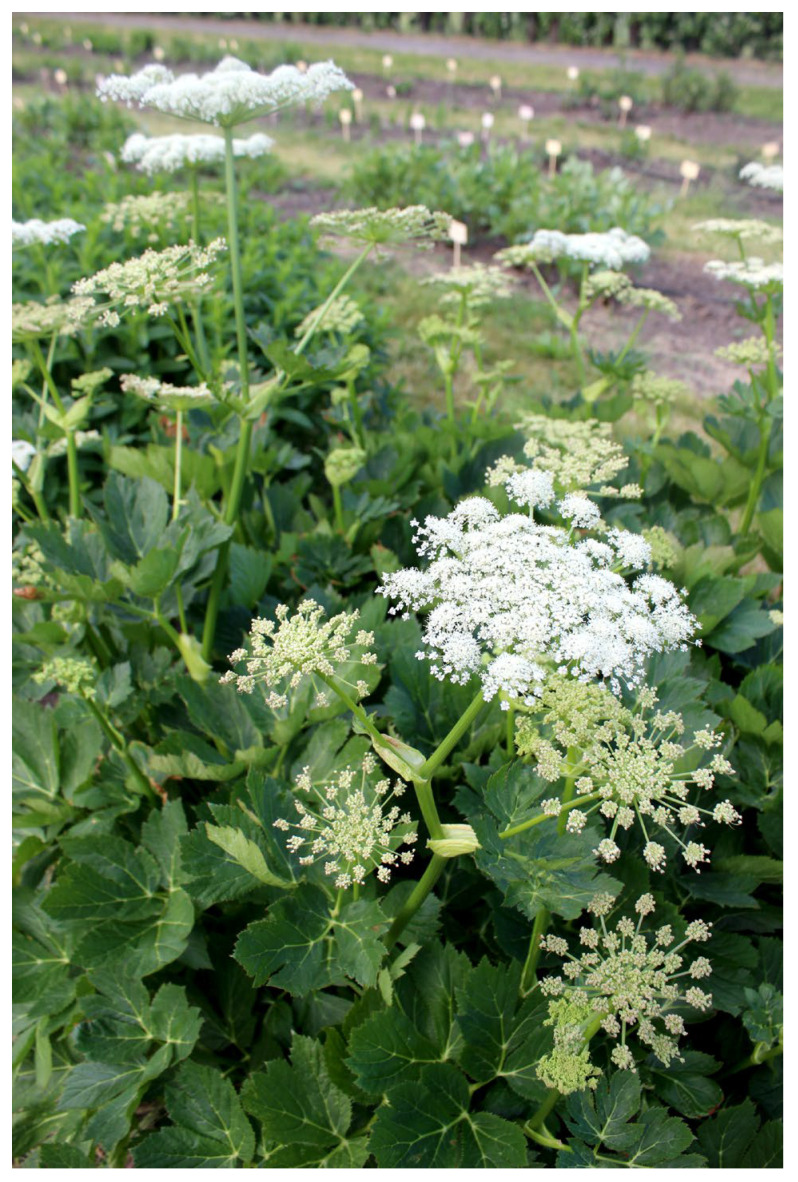
*Peucedanum ostruthium* ‘Kobota’. One of the samples of masterwort collected, evaluated and described in Swedish nationwide inventories of cultivated plants (Photo: Erik de Vahl).

**Figure 5 plants-12-00116-f005:**
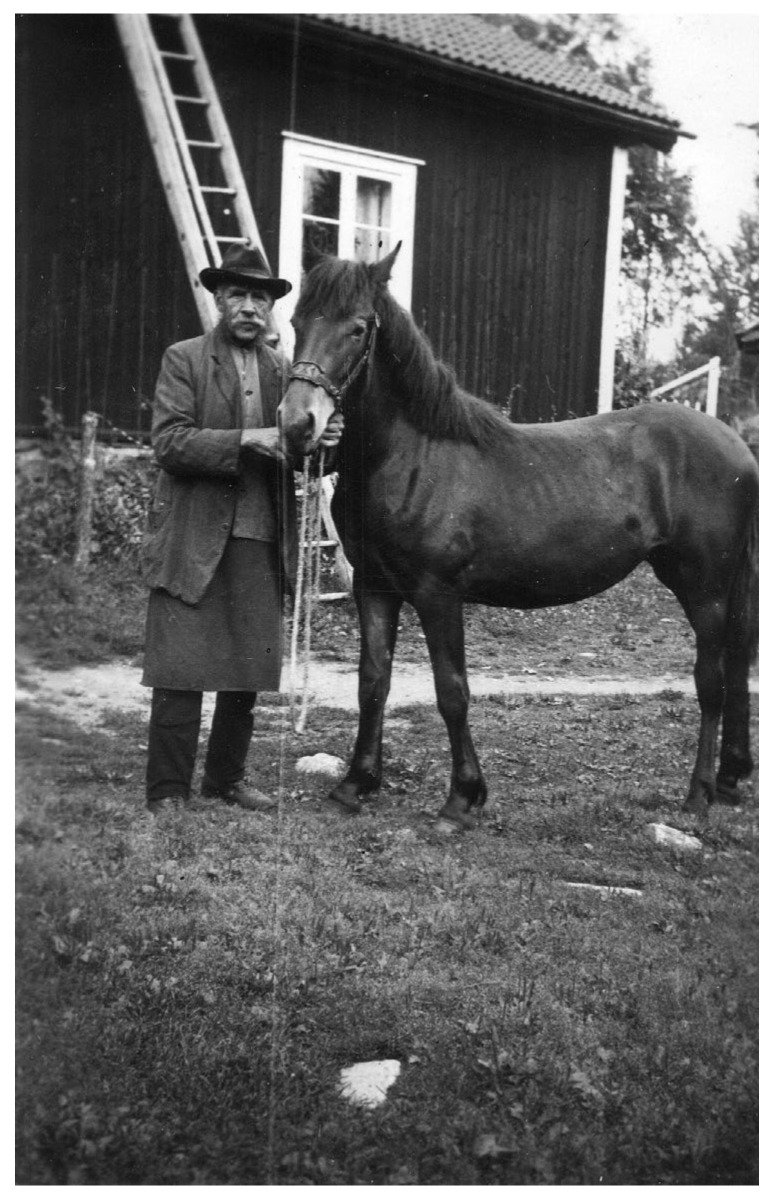
Rällsjö Anders Jansson (1863–1951) used masterwort in a remedy for horses called ‘Kampkur’ (horse remedy). Many plants from his garden in Dalecarlia are preserved in the Swedish National Gene Bank (Photo from private collection).

**Figure 6 plants-12-00116-f006:**
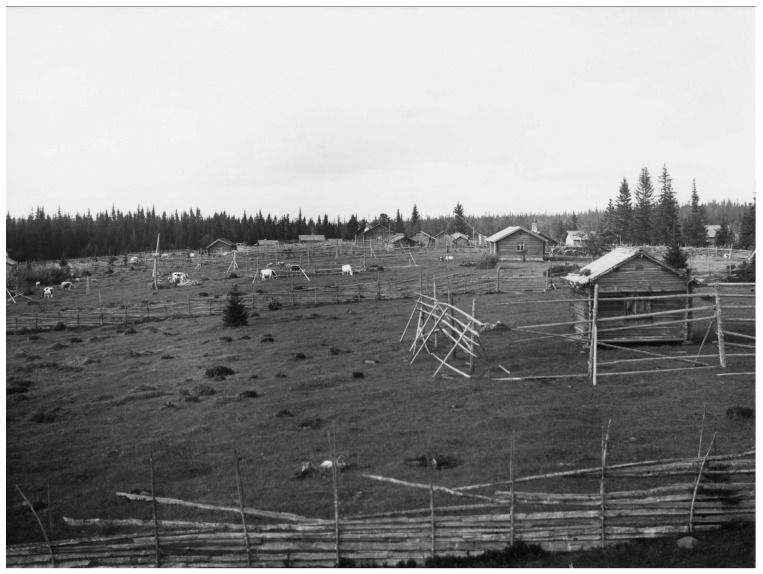
Pasture at the shieling Rismyren, Lima parish, Dalecarlia, in the early twentieth century (Photo Georg Renström, Courtesy: The Nordic Museum, Stockholm, NMA.0038914).

## Data Availability

All data generated or analysed during this study are available in this article.

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
