# Peer review of "“Cow Healers Use It for Both Horses and Cattle”: The Rise and Fall of the Ethnoveterinary Use of Peucedanum ostruthium (L.) Koch (fam. Apiaceae) in Sweden"

_plants, 2022, doi:10.3390/plants12010116_

Round 1

Reviewer 1 Report

Title: What is the origin of the quote “Cow healers use it for both horses and cattle”? I recommend referencing it in the text of the article.

Line 53: Consider revising the plant life form ("deciduous hemicryptophyte and stem plant") according to Ellenberg & Mueller-Dombois (1965-1966), available at <https://www.e-periodica.ch/cntmng?pid=bgi-002%3A1965%3A37%3A%3A130>.

Lines 57-58: Consider mentioning the compound umbels, with flowers sit in umbellules (elementary umbels), each with 30-45 flowers, arranged in larger (composed) umbels, each with 30-45(60) rays (radius). See, e.g., <http://www.floraiberica.es/floraiberica/texto/pdfs/10_129_74%20Peucedanum.pdf>.

Line 65: Lower altitudinal range can go down to 800 m. See, e.g., <http://www.floraiberica.es/floraiberica/texto/pdfs/10_129_74%20Peucedanum.pdf>.

Line 68: Consider replacing "isolated occurrences" by "scattered occurrence".

Line 77: I recommend replacing "find sites" by a clearer expression, e.g., "recording sites".

Line 90: Figure 2: I suggest the inclusion of a diagram showing Sweden's location in Europe. I strongly recommend indicating latitude and longitude on the distribution map.

Lines 93-97: Is this information a result of personal observation? Or does it also come from literature? I recommend that this should be clarified. Also, consider replacing "other than at botanical gardens" by "notably in botanical gardens".

Line 145: Consider replacing "has change" by "has changed".

Line 174-175: Consider replacing by "The Italian physician Pietro Andrea Mattioli propagated it for its usefulness in making remedies..."

Lines 181-182: Caption of Figure 3: I recommend that the source (repository) of the image is mentioned.

Lines 199-200: I strongly recommend that you provide an example of such data, to illustrate the continued use of the plant.

Line 235: Caption of Figure 5: Consider replacing "Private photo" by "Photo from private collection".

Line 242-243: I suggest that you mention who made such sensory tests.

Lines 268-269: I strongly recommend that the bibliographic reference of the quote be placed right at the end of this sentence.

Line 291: "plant material's cultivation history": Do you mean "material history of plant cultivation"? I suggest a clarification.

Line 292: Consider replacing "older" by "ancient".

Line 293: "lost their function and place in gardens": Avoid redundancy with the same expression (lines 277-278).

Line 351: Delete the extra full stop at the end of the bibliographic reference.

Line 373: Format the genus name in italics.

Author Response

We are grateful to reviewer # 1 for suggesting several changes which are improving the text. We have followed almost all of them.

Reviewer 1 Title: What is the origin of the quote “Cow healers use it for both horses and cattle”? I recommend referencing it in the text of the article. CLARIFIED IN LINE 261

Line 53: Consider revising the plant life form ("deciduous hemicryptophyte and stem plant") according to Ellenberg & Mueller-Dombois (1965-1966), available at <https://www.e-periodica.ch/cntmng?pid=bgi-002%3A1965%3A37%3A%3A130>. WE HAVE SIMPLIFIED THE SENTENSE

Lines 57-58: Consider mentioning the compound umbels, with flowers sit in umbellules (elementary umbels), each with 30-45 flowers, arranged in larger (composed) umbels, each with 30-45(60) rays (radius). See, e.g., <http://www.floraiberica.es/floraiberica/texto/pdfs/10_129_74%20Peucedanum.pdf>. WE HAVE CONSIDERED THIS BUT FIND IT NOT NECESSARY IN AN ETHNOBOTANICAL STUDY, WE HAVE GOOD PICTURES OF THE COMPOUND UMBELS IN FIG 1 AND 4

Line 65: Lower altitudinal range can go down to 800 m. See, e.g., <http://www.floraiberica.es/floraiberica/texto/pdfs/10_129_74%20Peucedanum.pdf>.  OK

Line 68: Consider replacing "isolated occurrences" by "scattered occurrence".  OK, WE HAVE CHANGED 

Line 77: I recommend replacing "find sites" by a clearer expression, e.g., "recording sites".  OK, WE HAVE REPLACED ACCORDING YOUR SUGGESTION

Line 90: Figure 2: I suggest the inclusion of a diagram showing Sweden's location in Europe. I strongly recommend indicating latitude and longitude on the distribution map. OK, WE HAVE PROVIDED A MAP OVER EUROPE SHOWING SWEDEN’S LOCATION IN EUROPE WHICH CAN BE PLACED BESIDE THE MAP OVER SWEDEN

Lines 93-97: Is this information a result of personal observation? Or does it also come from literature? I recommend that this should be clarified. Also, consider replacing "other than at botanical gardens" by " notably in botanical gardens ". WE HAVE CLARIFIED FROM WHERE THE INFORMATION COME.

Line 145: Consider replacing "has change" by "has changed". OK, WE HAVE CORRECTED

Line 174-175: Consider replacing by "The Italian physician Pietro Andrea Mattioli propagated it for its usefulness in making remedies..."  OK, WE HAVE REFRASED THE SENTENSE 

Lines 181-182: Caption of Figure 3: I recommend that the source (repository) of the image is mentioned. THE SOURCE OF THE ILLUSTRATION IS ALREADY MENTIONED IN THE CAPTION, NOT NECESSARY TO ADD FURTHER INFORMATION

Lines 199-200: I strongly recommend that you provide an example of such data, to illustrate the continued use of the plant. OK FOR FURTHER EXAMPLES SEE ALSO FOR INSTANCE LINE 255-256

Line 235: Caption of Figure 5: Consider replacing "Private photo" by "Photo from private collection". OK, DONE

Line 242-243: I suggest that you mention who made such sensory tests. OK CLARIFIED

Lines 268-269: I strongly recommend that the bibliographic reference of the quote be placed right at the end of this sentence.  OK, DONE

Line 291: "plant material's cultivation history": Do you mean "material history of plant cultivation"? I suggest a clarification.  WE HAVE REFRASED (cultivation history of a plant)

Line 292: Consider replacing "older" by "ancient". NO, I THINK OLDER IS BETTER HERE

Line 293: "lost their function and place in gardens": Avoid redundancy with the same expression (lines 277-278). CHANGED

Line 351: Delete the extra full stop at the end of the bibliographic reference. OK, DONE

Line 373: Format the genus name in italics. OK, DONE

Reviewer 2 Report

The  Article  “Cow healers use it for both horses and cattle”: The rise and  fall of the ethnoveterinary use of Peucedanum ostruthium (L.) Koch (fam. Apiaceae) in Sweden presents a cultural relict plant of an intentionally introduced species known in Sweden as early as the Middle Ages. It firs well to the topic of the special issue "Historical Ethnobotany: Interpreting the Old Records". The article is well-written and scientifically sound. It presents valuable information and the analyses are well done. It can be accepted as it is.

Author Response

We wish to thank Reviewer # 2 for the kind words.